# Dynamics of Vegetation Productivity in Relation to Surface Meteorological Factors in the Altay Mountains in Northwest China

**Aishajiang Aili** 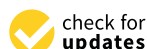**, Hailiang Xu \*, Xinfeng Zhao, Peng Zhang and Ruiqiang Yang**

State Key Laboratory of Desert and Oasis Ecology, Xinjiang Institute of Ecology and Geography,
Chinese Academy of Sciences, Urumqi 830011, China
\* Correspondence: xuhl@ms.xjb.ac.cn

**Abstract:** Vegetation productivity, as the basis of the material cycle and energy flow in an ecosystem, directly reflects the information of vegetation change. At the ecosystem level, the gross primary productivity (GPP) refers to the amount of organic carbon fixed by plant bodies. How to accurately estimate the spatiotemporal variation of vegetation productivity of the forest ecosystem in the Altay Mountains in northwest China has become a critical issue to be addressed. The Altay Mountains, with rich forest resources, are located in a semi-arid climate zone and are sensitive to global climate changes, which will inevitably have serious impacts on the function and structure of forest ecosystems in northwest China. In this paper, to reveal the variation trends of vegetation gross primary productivity (GPP) and its response to surface meteorological factors in the Altay Mountains in northwest China, daily temperature and precipitation data from the period of 2000–2017 were collected from seven meteorological stations in Altay prefecture and its surrounding areas; the data were analyzed by using the MODIS GPP model, moving average trend analysis, linear regression analysis and the climate tendency rate method. The results show that: (1) The spatial distribution pattern of GPP in the whole year was almost the same as that in the growing season of vegetation in the Altay Mountains. In the whole mountain range, the proportion of the area which had a GPP value of 400–600 g c/m$^2$ had the highest value; the proportion of the annual and growing season of this area was 41.10% and 40.88%, respectively, which was mainly distributed in the middle and west alpine areas of the Altay Mountains. (2) There was a big gap in the GPP value in the different stages of the vegetation growing season (April to September), which reached the highest value in July, the area with a GPP of 100–150 g c/m$^2$ was the highest, with 36.15%. (3) The GPP of the Altay Mountains showed an overall increasing trend, but the annual fluctuation was relatively large. In 2003, 2008, 2009 and 2014, the GPP showed lower values, which were 385.18 g c/m$^2$, 384.90 g c/m$^2$, 384.49 g c/m$^2$ and 393.10 g c/m$^2$, respectively. In 2007, 2011 and 2016, the GPP showed higher values, which were 428.49 g c/m$^2$, 428.18 g c/m$^2$ and 446.61 g c/m$^2$. (4) In 64.85% of the area of the Altay Mountains, the GPP was positively correlated with annual average temperature, and in 36.56% of the area, the correlation coefficient between temperature and GPP ranged from −0.2 to 0. In 71.61% of the area of the Altay Mountains, the GPP was positively correlated with annual accumulated precipitation, and in 28.39% of the area, the GPP was negatively correlated with annual accumulated precipitation. Under the scenario of global climate change, our study has quantitatively analyzed the long-term dynamics of vegetation GPP and its responses to meteorological factors in the Altay Mountains, which would be helpful for evaluating and estimating the variation trends of forest ecosystems in China, and has important guiding significance for policy formulation to protect forest resources and improve the local ecological environment.

**Keywords:** GPP; temporal and spatial change; climate tendency rate method; meteorological factors; Altay Mountains



## 1. Introduction

Climate change is an important factor affecting forest growth. Therefore, approaching the impacts of climate change on forest ecosystems is of great significance to ameliorate degraded land and support forestry development. Gross primary productivity (GPP) refers to the amount of organic carbon fixed by organisms, mainly by green plants, through photosynthesis in a unit of time, also known as total primary productivity [1–4]. The GPP determines the initial amounts of material and energy entering the terrestrial ecosystem [5–8]. Climate is the main factor that determines the distribution of forests and other species. Temperature and precipitation are the two most significant climate factors that affect the characteristics and distribution of forest ecosystems [6–11]. Forest resources in the Altay Mountains are the key component of the Xinjiang terrestrial ecosystem and are a green reservoir for regulating and conserving water resources [12–14]. Therefore, revealing the internal relationship between climate change and vegetation productivity in the Altay Mountains has important practical significance for improving the quality and productivity of forest land in the Altay Mountains, improving the management level of forest land, and promoting the healthy development of forest land [15,16]. Due to the close relationship between forests and climate, climate change will inevitably have a certain impact on forests [17–19].

The accurate estimation of the gross primary production of vegetation is vital for understanding the global carbon cycle and predicting future climate change. Multiple GPP products are currently available based on different methods and research tools, but their performances vary substantially when validated against GPP estimates from eddy covariance data. Measuring or estimating vegetation productivity mainly includes direct harvesting, volume conversion, and biomass equation methods. The harvesting method was developed first, where the biomass of each component is directly obtained by cutting and weighing all trees or estimating the biomass by directly measuring the biomass of the standard trees [20]. Direct harvesting requires a lot of human and material resources while causing unavoidable damage to the forest and the environment. The volume conversion method can be conducted due to the significant correlation between the stand volume and biomass. Nevertheless, the output of this model cannot provide an accurate estimation of forest biomass as it is related to forest density, age and site conditions. Biomass growth models can describe changes in an individual tree or stand size over time. Although these methods are widely used to connect individual trees and stand levels [21,22], the errors in estimating the stand biomass using the individual tree biomass growth combined with the stand structure are almost not quantified.

Compared with other research tools, the MODIS data can effectively reflect the landscape information on a regional scale. The MODIS-GPP product (MOD-17) is a global GPP product using remote sensing data developed according to the light-use efficiency principle. The GPP product has been widely used in appraisal and application research on various vegetation productions [23].

Research on the impact of climate change on forest ecosystems began in 1990 in China. However, most of this research work focused mainly on the variation of the forest ecosystem and its response to the annual average change in climate indicators, and little or no consideration was given to the relationship between seasonal changes in meteorological factors and vegetation productivity [23–25]. Zhou et al. analyzed the vegetation vitality in northern Eurasia and northern North America by using satellite data from 1981 to 1999; the analysis indicated that the vegetation vitality increased significantly and the growth period was prolonged [26]. Wang Yingying analyzed the temporal and spatial changes in vegetation phenology and their impact on GPP in temperate regions of China by using MODIS data [27]. Yu Xiaozhou established the estimation equation of the daytime dark respiration of different tree species according to the characteristics of dark respiration and light inhibition intensity of a single leaf from each tree species, and revised the GPP estimation results of a broad-leaved Korean pine forest on Changbai Mountain in combination with the measurement results of the canopy leaf biomass [28]. The increase in

temperature increases the NPP of the cold zone or subalpine forest ecosystem, and increases the decomposition rate and reduces the NEP of the forest ecosystem [29–31]. Cao et al. showed that the productivity of China's terrestrial ecosystem was highly sensitive to climate change from 1981 to 2000, and the interannual changes in NPP were significantly positively correlated with the temperature [32].

Due to the unique characteristics of the alpine climate, arid–semi-arid environment, and distinct vertical zonality, the vegetation productivity in forest ecosystems in the Altay Mountains are very sensitive to climate change and human disturbances. Therefore, an accurate assessment of the long-term dynamics of GPP in forest ecosystems in the context of global change and its mechanistic analysis of interannual variability would be helpful to estimate and predict the variation of Chinese forest ecosystems, and provide important guidance for policy formulation on the protection of forest resources. The variation of vegetation GPP in forest ecosystems is affected by complex interactions between climate, topography, forest structure, and soil fertilities. Among them, precipitation and temperature are determining factors affecting the vegetation GPP. However, detailed studies that take into account the decisive role of these factors are scarce. This study, therefore, was conducted to partly fill in this information gap. The MODIS-GPP product is a global GPP product using remote sensing data, developed according to the light-use efficiency principle. The MODIS data can effectively reflect the landscape information on a regional scale. To our knowledge, this study is the first research that analyzes the relationship between vegetation GPP and meteorological factors in the Altay Mountains area. In view of the significant decline in the ecological quality of the Altay Mountains in recent years and the unclear impact of climate change on the ecosystem, this study uses the MODIS GPP model, moving average trend analysis, linear regression analysis, and climate tendency rate method to reveal the spatial–temporal variation of vegetation productivity in the Altay Mountains and its response to meteorological factors, such as temperature and precipitation, by the means of field monitoring, computer simulation, and remote sensing technology. By combining the study with meteorological data and field observation data, we are able to accurately evaluate the variation trends of vegetation GPP in relation to meteorological factors by using the MODIS-GPP model. The methods used in this study to evaluate the variation trends of vegetation GPP can be used to assess ecosystem degradation, design the key restoration areas, as well as mitigate the impact of various meteorological disasters on mountain forest ecosystems. Due to the unavailability of systematic data on the impact of natural disasters and human disturbance, we are not able to carry out systematic research on the reasons of degradation of the forest ecosystem in the Altay Mountains. Further research can be conducted by combining the study with field investigations and plant physiology experiments on the basis of a longer period of remote sensing data.

## 2. Material and Methods

### 2.1. Description of Study Area

The Altay Mountains spans China, Kazakhstan, Russia and Mongolia, with a total length of approximately 2000 km. It runs from northwest to southeast. In this study, the Altay Mountains in China is selected as the study area. This area belongs to the south slope of the middle section of the Altay Mountains, with a length of approximately 450 km from east to west and a width of approximately 80–150 km from north to the south. The mountain gradually becomes narrow from northwest to southeast, showing the topographic characteristics of high and wide in the northwest and low and narrow in the southeast (Figure 1), with a total area of $2.6 \times 10^4$ km$^2$ [33].

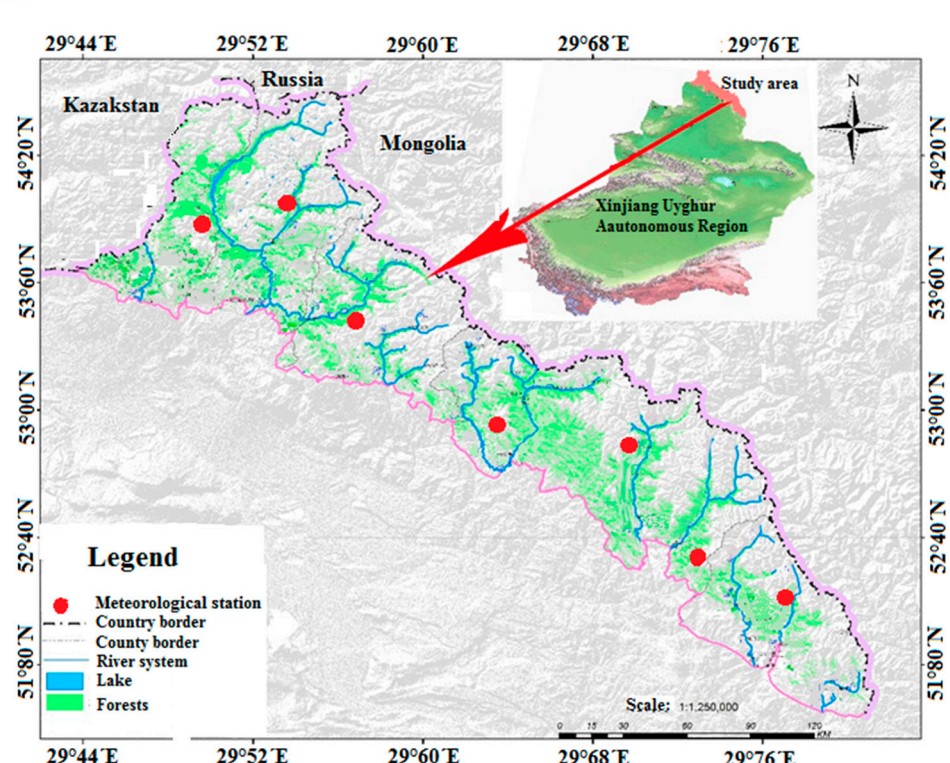

**Figure 1.** Location of Study Area (Driving number of map: GS (2019) 1823).

The Altay Mountains has a vast area of forests, grasslands and abundant wetlands, which are important components to maintain the ecological functions of the mountain area. Among them, the forest area is 9802 km$^2$, accounting for 37.7%, and the total area of the wetland is approximately 800 km$^2$, accounting for 3% of the total area of the Altay Mountains. The forest is mainly composed of Siberian larch (*Larix sibirica*), and the main arbors include Siberian spruce (*Picea obovata*) and Betula pendula, which, respectively, constitute the larch forest, spruce forest, larch–spruce mixed forest and Betula pendula mixed forest. There is a large number of peat swamps. It is the largest and most complete peat swamp and the key "carbon pool" for water conservation in Xinjiang.

The Altay Mountains, with continental climate characteristics, are located in the hinterland of Eurasia and far from the sea. The spring temperature rises quickly and is windy, the summer is cool and short, the autumn temperature drops quickly and is sunny, and the winter is cold and long. The annual average temperature is approximately −2 °C, and the extreme maximum temperature is 33.3 °C. The mountain area with an altitude of 3100~3300 m is covered with snow all year. The annual average temperature in the middle- and high-mountain area, with an altitude of 1400~2600 m, is below −9 °C, and is the hottest in July, which is only approximately 15 °C. The annual temperature difference is approximately 30 °C, and the daily temperature difference is approximately 12 °C. The annual average temperature in the low-mountain and hilly area is below 4 °C. The altitude increases by 100 m, and the annual precipitation increases by 30/80 mm. The annual precipitation is 200~300 mm in the low-mountain belt, 300~500 mm in the middle-mountain belt, and 600~800 mm in the high-mountain belt, and decreases from north to south and from west to east. The Altay Mountains are an important natural forest area in Xinjiang, known as the ecological protective screen in the north of Xinjiang. The total area of the forest is 69 × 10$^4$ hm$^2$, and the standing forest volume is 9.197 × 10$^4$ m$^3$, accounting for 47% of the natural forest in Xinjiang [14,34,35].

*2.2. Data Sources*

The daily meteorological data of seven meteorological stations around the Altay Mountains (Altay Meteorological Bureau, Burqin County Meteorological Bureau, Habahe County Meteorological Bureau, Jimunai County Meteorological Bureau, Fuhai County Meteorological Bureau, Fuyun County Meteorological Bureau, Qinghe County Meteorological Bureau) from 2000 to 2017 were obtained from China Meteorological Data Sharing Service Network (http://data.cma.gov.cn/, 12 October 2018), and the meteorological data were strictly controlled to maintain the accuracy of the research results. To ensure the accuracy of meteorological data, the inverse distance weighted (IDW) method was used to interpolate the site data of the meteorological data. The selected meteorological stations were evenly distributed in the whole study area. The daily average temperature and precipitation data were used to reveal the relationship between meteorological factors and vegetation GPP. The NDVI (Normalized Differentiated Vegetation Index) data used in this paper were the MODIS MOD13Q1 products, with a spatial resolution of 250 m and a time series from January 2000 to December 2017. First, the obtained remote sensing data were preprocessed by data format conversion, mosaic, projection conversion and study area extraction. Further, in order to reduce the impact of noise information on the data, Savitzky Golay filtering [36] and MVC synthesis processing [37] were also performed on the NDVI data to obtain annual NDVI data representing the best condition of vegetation growth. Savitzky Golay filtering was used in this study because of its analytical and computational simplicity, and good smoothing capabilities. The MVC (Maximum Value Composite) is similar to that used in the AVHRR-NDVI product, whereby the pixel observation with the highest NDVI value is selected to represent the entire period. The MOD13Q1 data used in this study were provided every 16 days at a 250 m spatial resolution as a gridded level 3 product. Furthermore, the vegetation indices were used for the global monitoring of vegetation conditions and were used for displaying the land cover changes. These data can be used for characterizing the biophysical properties of land surface, including primary production and land cover conversion [36,37].

*2.3. Statistical Methods*

2.3.1. Climate Tendency Method

Using the time series of meteorological factors, with time as the independent variable and meteorological factors as the dependent variable, set $Y$ as the meteorological variable and $t$ as the time, and establish a linear regression equation between $Y$ and $t$. The Pearson correlation coefficient was calculated to obtain the linear relationship between vegetation GPP and meteorological factors. The climate tendency rate of meteorological factors is expressed by the following formula:

$$Y_i = a_0 + a_1 t_i \tag{1}$$

where $Y_i$ is meteorological factor, $t_i$ is time, $\alpha_1$ is linear trend term, $\alpha_1 \times 10$ is the climate tendency rate of meteorological elements every 10 years, with the unit of 10 years. When it is less than 0, it means that the meteorological factor sequence decreases with time, otherwise it increases. The larger the absolute value of $\alpha_1$, the more significant the trend [38–41].

2.3.2. Correlation Analysis

In this study, the dimensionless climate trend coefficient $r_{xt}$ is obtained by using the following equation:

$$r_{xt} = \frac{\sum_{t=1}^{n} (x_t - x)(t - \frac{n+1}{2})}{\sqrt{\sum_{t=1}^{n} (x_t - x)^2 (t - \frac{n+1}{2})^2}} \tag{2}$$

In the equation, $r_{xt}$ is the trend coefficient, and its significance can be tested by $t$ distribution statistics. The relationship between tendency rate b and trend coefficient $r_{xt}$ is as follows:

$$b = r_{xt}(\sigma_x/\sigma_t) \tag{3}$$

where $\sigma_x$ and $\sigma_t$ are the mean square deviations of factor sequence and natural sequence, respectively.

The Pearson correlation coefficient (R) is employed to present the relationship of vegetation productivity with precipitation and temperature. A high R-value indicates a strong correlation, and a low R-value represents a weaker correlation. The correlation analysis for each grid is performed through Matlab software by writing code programs.

### 2.3.3. GPP Inversion of Total Primary Productivity

In this study, the GPP inversion of total primary productivity is obtained by using the MODIS GPP model. The precise quantification of GPP at the landscape level is however challenging as there is no direct measurement technique beyond the leaf level. Forest productivity is primarily estimated using the data gathered through temporal forest inventories and established empirical allometric equations in terms of biomass. The MODIS GPP model is established based on the linear relationship between GPP and the photosynthetically active radiation absorbed by vegetation. The calculation process is as follows:

$$GPP = APAR \times \varepsilon_{\max} \times f(a\min) \times f(VPD) \tag{4}$$

$$APAR = SWRad \times 0.45 \times (1 - e^{k \times LAI}) \tag{5}$$

where GPP is the total primary productivity, the unit is g c·m$^{-2}$·s$^{-1}$; $\varepsilon_{Max}$ is the maximum light energy utilization rate, the unit is kg C/MJ. The light energy utilization rate ($E_{max}$) is the ratio of the energy stored to the energy of light absorbed. The amount of energy stored can only be estimated because many products are formed, and they vary with the plant species and environmental conditions. APAR (Absorbed Photosynthetic Active Radiation) is the photosynthetically active radiation absorbed by vegetation, and the unit is MJm$^{-2}$·s$^{-1}$, thus, representing the product with a 45% incident short-wave radiation (SWRad) and a photosynthetically active radiation ratio which is absorbed by the vegetation canopy. The SWRad is radiation at wavelengths shorter than 4 microns. Usually, radiation is in the visible and near-infrared wavelengths. This radiation ratio is calculated using LAI (Leaf Area Index) through simple Beer's law; this study used the GlobMap LAI data from 2000 to 2017. GlobMapLAI is a global long-term series LAI product generated based on AVHRR/MODIS data. $k$ is the canopy extinction coefficient—generally, its value is 0.5. $f$ (VPD) and $f$ ($_a$min) are the correction factors of the vapor pressure difference and air temperature at a 2 m height—they are calculated using the following equation:

$$f(VPD) = \frac{VPD_{max} - VPD}{VPD_{max} - VPD_{min}} \tag{6}$$

$$f(T_{min}) = \frac{T_{min} - T_{min\_min}}{T_{min\_max} - T_{min\_min}} \tag{7}$$

where $VPD_{max}$ and $T_{min\_max}$ are the maximum daily vapor pressure difference (Pa) and maximum daily air temperature (°C) at the time of maximum photosynthetic efficiency, respectively; $VPD_{min}$ and $T_{min\_min}$ are the minimum vapor pressure difference (Pa) and daily minimum temperature (°C) when photosynthesis is 0, respectively. These parameters are default parameters in the BPLUT table [42–44]. In this study, the 500 m MODIS reflectance data (MOD09GA) were used for spectral unmixing. The MOD09GA product was provided along with daily 500 m spatial resolution surface reflectance data, which were generated from Terra-MODIS bands 1 to 7 (620 nm–2155 nm). The MODIS data collection and resolution methods used in this study are available and well explained in the

website (https://modis-images.gsfc.nasa.gov/_docs/CMUSERSGUIDE.pdf, 12 October 2018), which is proposed by Kathleen in 2012 [45].

## 3. Results and Discussions

### 3.1. General Characteristics of Vegetation Productivity in the Altay Mountains

In this study, both the annual GPP and growing season GPP of vegetation in different parts of the Altay Mountains were compared (Figure 2).

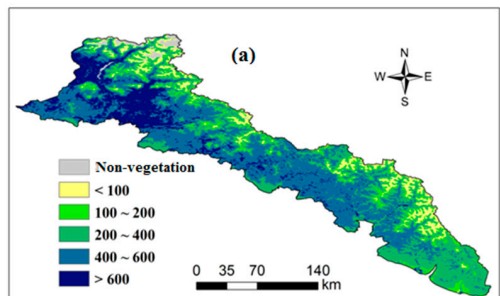 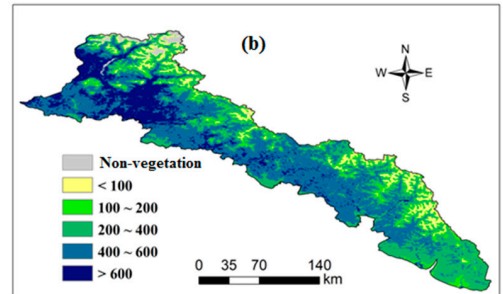

**Figure 2.** Overall characteristics of annual GPP (**a**) and growing season GPP (**b**) in the Altay Mountains.

It can be seen from Figure 2 that the spatial distribution pattern of annual GPP and growth season GPP of vegetation in the Altay Mountains is almost the same, and the proportion of each grade is also very similar. In the whole mountain area, the area with a GPP of 400~600 g c/m$^2$ has the highest proportion, with annual and growing season proportions of 41.10% and 40.88%, respectively (Table 1). These areas are mainly distributed in the middle- and high-mountain areas in the middle and west parts of the Altay Mountains. The area with a GPP of 200~400 g c/m$^2$ ranks the second, with annual and growing season GPP proportions of 29.21% and 29.96%, respectively, which are mainly distributed in the areas with a higher altitude in the east and west parts of the Altay Mountains. The area with a GPP of <100 g c/m$^2$ is the smallest, with annual and growing season GPP proportions of 6.17% and 6.28%, respectively, which are mainly distributed in high-altitude areas in the west and east parts of the Altay Mountains.

**Table 1.** Proportion of the area with a different GPP in annual and growing season.

| Period | GPP (g c·m$^{-2}$) | | | | |
|---|---|---|---|---|---|
| | <100 | 100~200 | 200~400 | 400~600 | >600 |
| Annual (%) | 6.17 | 9.18 | 29.21 | 41.10 | 14.34 |
| Growing season (%) | 6.28 | 9.26 | 29.96 | 40.88 | 13.62 |

The GPP of different months in the growing season is significantly different. In this study, the proportion of the area with a different GPP from April to September was compared, and the distributions of the higher GPP area and lower GPP area were analyzed (Figure 3).

It can be seen from Figure 3 that, in April, there is still a large area of snow cover in the high-mountain area. In the areas with vegetation coverage, the proportion of the area with a GPP of 1~5 g c/m$^2$ is the highest, which is 36.73%, and is mainly distributed in the areas below the snow line in the middle part of the mountain. The GPP with 83.73% of the area is less than 15 g c/m$^2$.

In May, the vegetation coverage area expanded and the vegetation GPP increased; the proportion of area with a GPP of 80~120 g c/m$^2$ is the largest, accounting for 39.04%, which was mainly distributed at the west part of middle- and lower-altitude area. However, there are still lower vegetation coverage areas in this month; the areas with a GPP < 10 g c/m$^2$ are mainly distributed under the snow line at a high altitude.

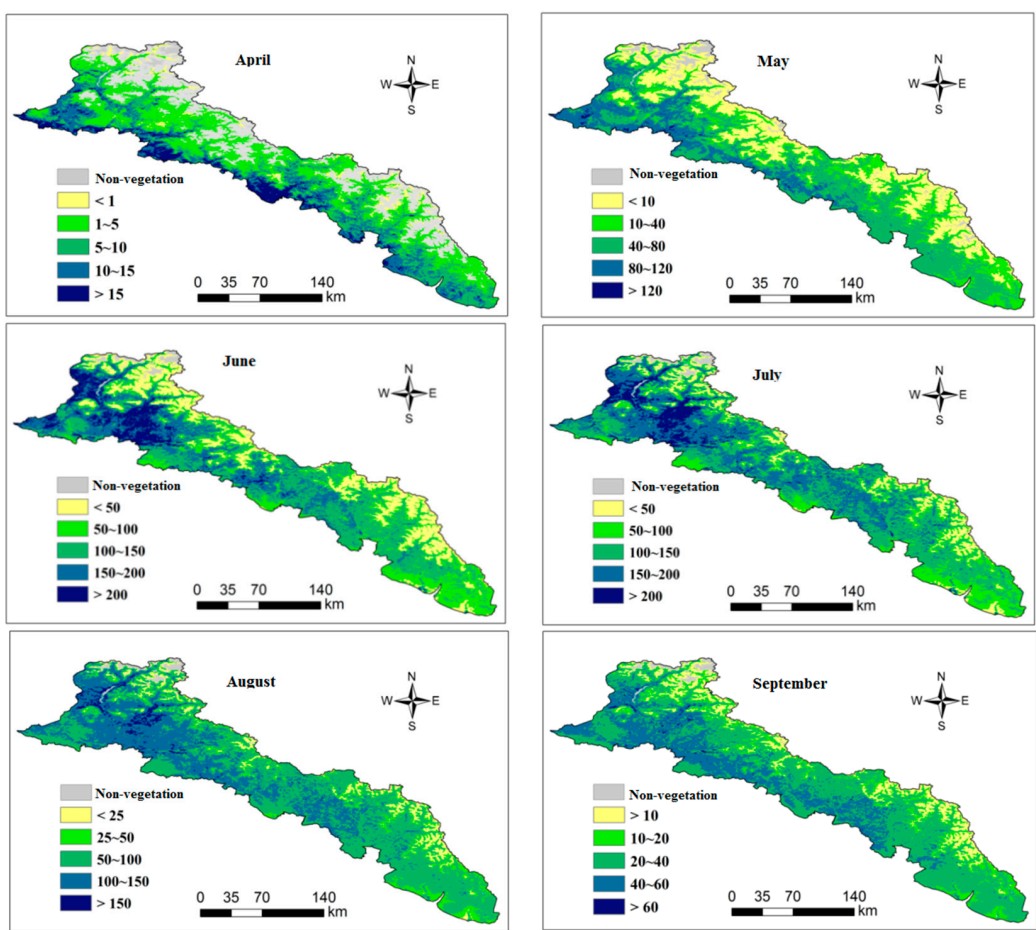

**Figure 3.** Distributions of GPP in the growing season in the Altay Mountains.

In June, the vegetation grew vigorously, and the vegetation GPP gradually increased. The proportion of area with a GPP of 100~150 g c/m$^2$ is the highest, which is 32.58%, and it is widely distributed in the east and middle parts of the mountain. The proportion of the highest vegetation coverage areas with a GPP > 200 g c/m$^2$ is 8.18%, which is mainly distributed in the western part of the mountain with a medium altitude.

In July, the vegetation GPP of the Altay Mountains was further improved, and the vegetation GPP of most parts of the mountain has a higher value and is relatively consistent. The proportion of area with a GPP of 100~150 g c/m$^2$ is the highest, which is 36.15%, and is mainly distributed at the east and middle parts of the mountain. In this month, there are still some lower vegetation coverage areas with a GPP < 50 g c/m$^2$, accounting for 8.75%, and are mainly distributed at the high-altitude areas in the east and west sections of the Altay Mountains.

In August, the GPP of vegetation in whole mountains decreased gradually, and the proportion of area with a GPP of 50~100 g c/m$^2$ is the highest, accounting for 49.48%, and is mainly distributed at the east and the middle parts of the mountain. The GPP of most parts of the mountain ranges from 50 g c/m$^2$ to 150 g c/m$^2$. However, there are still some lower-vegetation coverage areas; the proportion of area with a GPP < 25 g c/m$^2$ accounts for 5.51% and is mainly distributed at the higher-altitude area in the west and east parts of the mountains.

In September, the GPP of whole mountain rapidly decreased; the area with a GPP of 20~40 g c/m$^2$ accounts for 51.38%. There were only small areas with a higher GPP; the proportion of area with a GPP > 60 g c/m$^2$ accounts for 0.65% (Table 2).

**Table 2.** The proportion of the areas with different GPP in the growing season in the Altay Mountains.

| April | | May | | June | | July | | August | | September | |
|---|---|---|---|---|---|---|---|---|---|---|---|
| GPP/g c·m$^{-2}$ | Proportion/% | GPP/g c·m$^{-2}$ | Proportion/% | GPP/g c·m$^{-2}$ | Proportion/% | GPP/g c·m$^{-2}$ | Proportion/% | GPP/g c·m$^{-2}$ | Proportion/% | GPP/g c·m$^{-2}$ | Proportion/% |
| <1 | 4.2 | <10 | 17.27 | <50 | 18.23 | <50 | 8.75 | <25 | 5.51 | <10 | 9.46 |
| 1~5 | 36.73 | 10~40 | 10.95 | 50~100 | 20.72 | 50~100 | 18.98 | 25~50 | 10.83 | 10~20 | 14.04 |
| 5~10 | 18.37 | 40~80 | 31.19 | 100~150 | 32.58 | 100~150 | 36.15 | 50~100 | 49.48 | 20~40 | 51.38 |
| 10~15 | 24.42 | 80~120 | 39.04 | 150~200 | 20.29 | 150~200 | 28.88 | 100~150 | 31.61 | 40~60 | 24.47 |
| >15 | 16.27 | >120 | 1.55 | >200 | 8.18 | >200 | 7.25 | >150 | 2.57 | >60 | 0.65 |

### 3.2. Annual Variation of Vegetation Productivity in the Altay Mountains

From 2000 to 2017, the change in vegetation GPP in the Altay Mountains fluctuated greatly, showing an increasing trend (Figure 4).

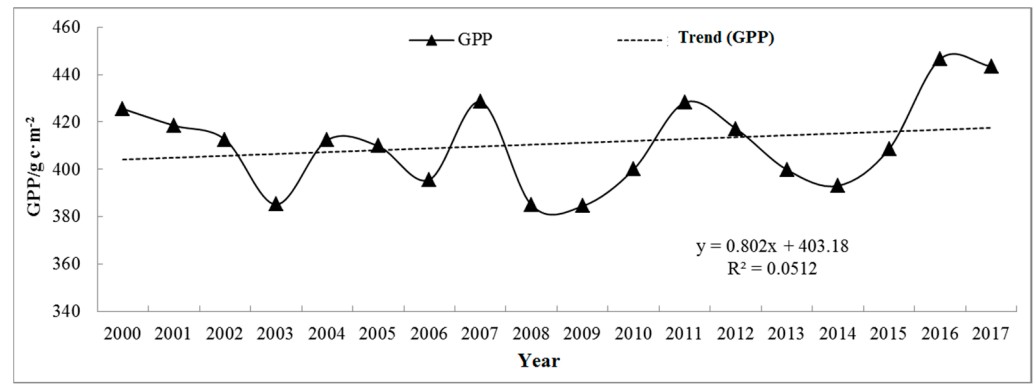

**Figure 4.** Annual variation of GPP in the Altay Mountains (2000–2017).

It can be seen from Figure 4 that the vegetation GPP of the Altay Mountains presents an overall increasing trend, but the interannual fluctuation is still large. The annual average value peaked in 2007, 2011 and 2016, and their values were 428.49 g c/m$^2$, 428.18 g c/m$^2$ and 446.61 g c/m$^2$. In 2003, 2008, 2009 and 2014, the values were relatively low, with 385.18 g c/m$^2$, 384.90 g c/m$^2$, 384.49 g c/m$^2$ and 393.10 g c/m$^2$, respectively.

In this study, the significance of variation trend of the vegetation GPP in the period of 2000–2017 was also analyzed, and the results are shown in Figure 5.

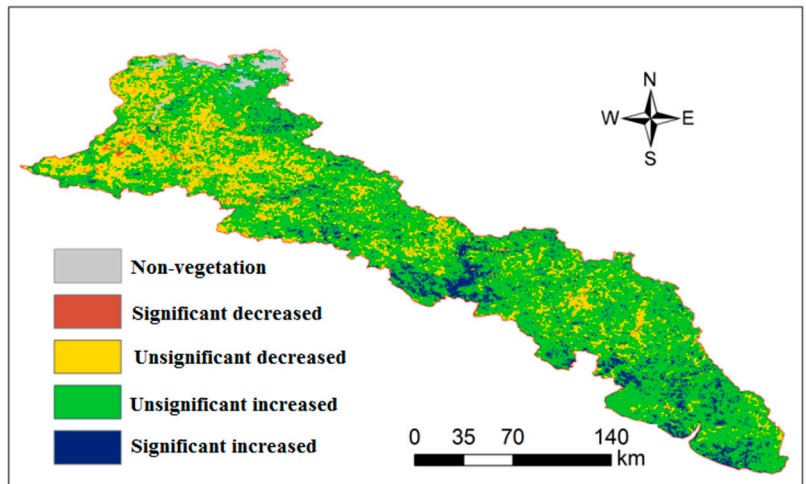

**Figure 5.** Significance of variation of GPP in the Altay Mountains.

It can be seen from Figure 5 that in 73.16% of the study area, the vegetation GPP has an increasing trend; the significantly increased area accounts for 7.72%, which is mainly distributed in the middle and eastern parts of the mountain. However, the vegetation GPP

shows a decreasing trend in 26.84% of the study area, and is mainly distributed at the western part of the mountain; the significantly decreased area accounts for 0.98%.

The results of the annual variation rate of the vegetation GPP during the study period are presented in Figure 6.

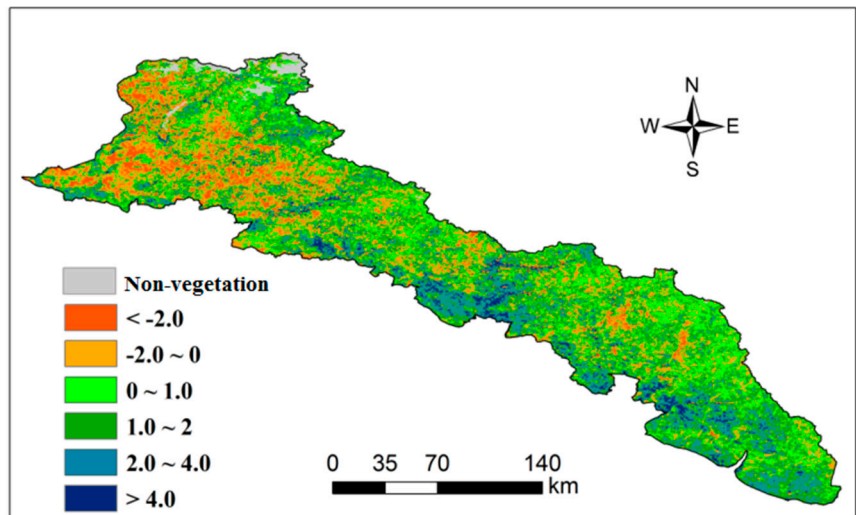

**Figure 6.** Variation rate of GPP in the Altay Mountains.

It can be seen from Figure 6 that the variation rates of the vegetation GPP in the study area are significantly different. The area with a GPP variation rate of $-2.0\sim0$ g·c·m$^{-2}$·a$^{-1}$, $0\sim1.0$ g·c·m$^{-2}$·a$^{-1}$ and $1.0\sim2.0$ g·c·m$^{-2}$·a$^{-1}$ accounts for 21.23%, 27.46% and 23.43%. The areas with a GPP variation rate of $-2.0\sim0$ g·c·m$^{-2}$·a$^{-1}$ are mainly distributed in the middle-height area of the western part of the Altay Mountains; the areas with a GPP variation rate of $0\sim1.0$ g·c·m$^{-2}$·a$^{-1}$ are mainly distributed in the western and eastern parts of the mountain; and the areas with a GPP variation rate of $1.0\sim2.0$ g·c·m$^{-2}$·a$^{-1}$ are distributed discretely in the whole part of the mountain. The areas with a GPP variation rate of $2.0\sim4.0$ g·c·m$^{-2}$·a$^{-1}$ account for 18.76%, and are mainly distributed at the lower-mountain area of the middle and east parts of the Altay Mountains. The areas with a GPP variation rate of $<-2.0$ g·c·m$^{-2}$·a$^{-1}$ account for 6.03%, and are mainly distributed in the middle-latitude area of the western section. The area with a GPP variation rate of $>4.0$ g·c·m$^{-2}$·a$^{-1}$ accounts only for 3.09%, which is scattered in the low-latitude areas in the middle and east sections.

### 3.3. Impact of Meteorological Factors on Vegetation GPP

The variation of the vegetation GPP of the forest is the outcome of the coupling influence of various factors including climate change, natural disasters and human disturbance. The changes in surface meteorological factors will inevitably affect the forest to a certain extent; the influence of temperature and precipitation on vegetation GPP is especially more significant. In this study, the impact of temperature and precipitation on vegetation GPP in different parts of the mountain were analyzed by comparing the correlation coefficient, which indicates the strength of relevance between them. Since the annual GPP and growing season GPP showed a similar distribution pattern, as mentioned above, the relationship between the annual average GPP in different parts of the mountain and meteorological factors were examined by using correlation analysis. However, the correlation between two independent variables, temperature and precipitation, was not considered because this was not relevant to our study. Temperature is the determining factor for the growth and development of vegetation. The stronger the impact of temperature on the vegetation GPP, the greater the correlation coefficient between them (Figure 7a).

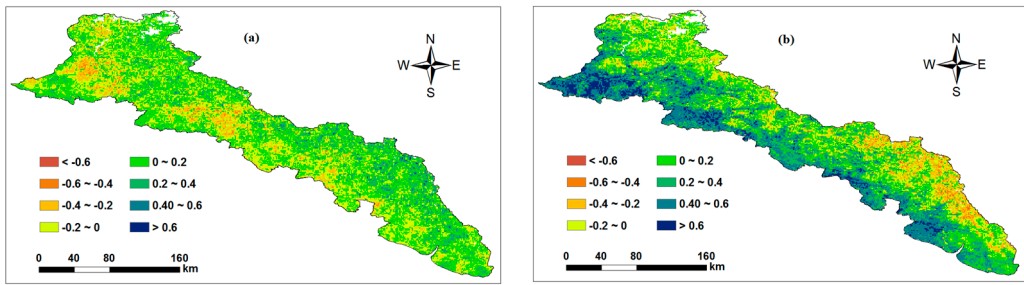

**Figure 7.** Correlation of vegetation GPP with temperature (**a**) and precipitation (**b**) in the Altay Mountains.

It can be seen from Figure 7a that vegetation GPP is positively correlated with the annual average temperature, with a mean correlation coefficient of 0.19. The area with positive correlations accounted for 64.85% of the whole, with a mean value of 0.29—this showed a wide distribution but was more intensive in the middle-altitude area of the southern slope of the Altay Mountains. The area with a correlation coefficient of 0–0.2 accounts for 30.23%, the area with a correlation coefficient of 0.2–0.4 accounts for 10.48%. However, in 35.15% of the study area, the correlation coefficient between temperature and the vegetation GPP has a negative value. The area with a correlation coefficient of −0.2–0 accounts for 13.2%. The areas with a negative correlation coefficient were mainly distributed in the north and southeast parts of the mountain with a lower altitude.

Similarly, precipitation also plays an essential role in vegetation GPP. It can be seen from Figure 7b that a GPP of 71.61% of the Altay Mountain area is positively correlated with the annual average precipitation, with a mean correlation coefficient of 0.39. Regarding the distribution of the area, it has a similar distribution pattern to temperature but a relatively wider distribution of the positively correlated areas. The areas with a correlation coefficient of 0–0.2, 0.2–0.4 and 0.4–0.6 account for 16.49%, 16.07% and 13.39%. However, a vegetation GPP of 28.39% of the study area is negatively correlated with the annual average precipitation, of which most were located in the north part of the mountain with a higher altitude.

Overall, the result of the correlation analysis indicates a strong linkage between temperature and precipitation with vegetation GPP in the study area. An increase in these variables will likely increase the vegetation productivity. This study's outcome aligns with some previous studies that established a connection between vegetation productivity and meteorological factors. Du et al. analyzed the influence of temperature and precipitation on forest ecosystems in the vegetation growing season in the Qilian Mountains of northwestern China, and found a significant interaction between temperature and precipitation, contributing up to 30% of total variability in the predicted ecosystem level in the vegetation growing season [46]. Meteorological factors determine the growth and development of vegetation in the Altay Mountains. Conversely, because the forest ecosystem is a huge carbon pool, it acts as a source or sink of $CO_2$ in the atmosphere, thus, further strengthening or offsetting future climate change. Benjamin et al. evaluates the sensitivity of forest productivity to precipitation and air temperature in inner Asian forests by using satellite remote sensing, dendrochronology and dynamic global vegetation model (DGVM) simulations, and indicated that in cool arid environments, precipitation is not the only limitation to forest productivity. Interactions between changes in precipitation and air temperature may enhance soil moisture stress while simultaneously extending the growing season length, with unclear consequences for net carbon uptake [47]. Wang et al. studied the impacts of climate change on forest growth in saline-alkali land of the Yellow River Deltain North China by using standard site methods and tree ring sampling, and indicated that precipitation is the main meteorological factor affecting tree growth, while temperature and air pressure are also significantly correlated with tree growth [48]. The Altay Mountains are located in the hinterland of Eurasia, far away from the sea, and have a few water vapor sources. In the situation of global atmospheric circulation, they are located in the westerly zone. The

westerly flow enters this area along the Ertish river valley, raising the condensation cloud to cause rain. Therefore, precipitation is relatively large, and has become one of the wetter regions in the Xinjiang. The variation trends of vegetation coverage in different parts of the mountain were quite different. In general, the areas where the vegetation coverage has decreased are mainly distributed in the west and middle parts of the Altay Mountains, the Piedmont of the mountains, as well as the plain areas in the southwest, central and northeast parts of the mountain.

## 4. Conclusions

The MODIS-GPP product is a global GPP product using remote sensing data, and is developed according to the light-use efficiency principle. The MODIS data can effectively reflect the landscape information on a regional scale. Combined with meteorological data and field observation data, the variation trends of vegetation GPP in relation to meteorological factors can be accurately evaluated by using the MODIS-GPP model.

The spatial distribution pattern of annual vegetation GPP and growing season GPP of the Altay Mountains was almost consistent. In the whole study area, the proportion of the areas with a vegetation GPP of 400~600 g c/m$^2$ was the highest, and was mainly distributed in the middle- and high-altitude areas of the middle and west parts of the Altay Mountains with the best plant growth conditions.

Due to the differences in meteorological factors such as temperature and precipitation in the Altay Mountains, the GPP in different stages of the vegetation growth season (April to September) had a great difference, reaching the highest value in July; the area with a GPP of 100~150 g c/m$^2$ had the highest proportion, with 36.15%. In April, there was still a large area of snow cover in the high-altitude area of the Altay Mountains, and the vegetation growth conditions were poor. Among the areas with vegetation coverage, the proportion of area with a GPP of 1~5 g c/m$^2$ was the highest, with 36.73%, and was mainly distributed in the areas below the snow line in the middle-altitude area.

The vegetation GPP of the Altay Mountains had an overall increasing trend in the study period, but there was a large gap in GPP values in different years—the peak value appeared in 2007, 2011 and 2016, and the lower values appeared in 2003, 2008 and 2009, respectively. In 73.16% of the study area, the vegetation GPP showed an increasing trend, and 7.72% of the area was in the middle and east parts of the mountain, this increasing trend was more significant. However, the decreasing trend appeared in 26.84% of the study area, and was mainly distributed in the west part of the mountain.

Temperature and precipitation were the most important meteorological factors affecting vegetation GPP in the Altay Mountains. In 71.61% of the area of the Altay Mountains, the vegetation GPP was positively correlated with precipitation, while the vegetation GPP in 64.85% of the study area was positively correlated with the annual average temperature.

Our study has quantitatively analyzed the long-term dynamics of vegetation GPP and its responses to meteorological factors in the Altay Mountains. The results of our study would be helpful in carrying out further research on the assessment of the degradation of the forest ecosystem, evaluation of the ecosystem function, identification of the key restoration areas, as well as mitigating the impact of various meteorological disasters on mountain forest ecosystems.

**Author Contributions:** Conceptualization, methodology, validation, formal analysis, investigation, A.A. and H.X.; Software visualization, A.A., P.Z., X.Z. and R.Y. All authors have read and agreed to the published version of the manuscript.

**Funding:** This study was jointly supported by the Natural Science Foundation of Xinjiang Uyghur Autonomous Region «Study on the Cooperative Configuration of Oasis Wind break and Sand fixation System in Extremely Arid Areas» and «Key technologies for natural forest protection and restoration in 2022 (ZX-2022040)».

**Data Availability Statement:** Data could be provided on reasonable request from the first author.

**Acknowledgments:** The authors are grateful to the staff of the Department of Forestry of Altay Prefecture for providing the necessary facilities during the study. We are also thankful to anonymous reviewers for their valuable suggestions and comments on the overall improvement of the manuscript.

**Conflicts of Interest:** The authors declare no conflict of interest.

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
