# Peer review of "Dynamics of Vegetation Productivity in Relation to Surface Meteorological Factors in the Altay Mountains in Northwest China"

_forests, doi:10.3390/f13111907_

Round 1
Reviewer 1 Report
1. Revise the introduction to make it more logically hierarchical.
Line-65: Literature [20-22] was published from 2008 to 2010. In fact, many related scholars have done this kind of research in recent years, which has little support as the introduction of scientific opinions and is not representative.
Line 84: It is mentioned here that this manuscript uses meteorological and ecological methods, as well as field monitoring techniques, which are not in fact represented in the manuscript.
2. Refine the overview of the research area.
Line-126: "The forest is mainly composed of Hana larch... ." This paper does not study GPP of different vegetation types in detail, so it is unnecessary to list it in such detail. If necessary, it should be placed in line-105 of the paper.
3. Modify data sources and research methods.
① Is it reasonable to use meteorological data from only 7 stations for analysis? What exactly are the meteorological data used?
② What method is used to interpolate the site data of meteorological data? Different difference methods will affect the spatial credibility of meteorological data.
③ The description of GPP model is not clear. What is the light energy efficiency (Emax) in the model? How was SWRad informed? How is LAI obtained?Line-138: What is the purpose of using MOD13Q1?
f (amin) in equation (4) does not agree with the statement in the following manuscript.
④ The method is too simple, and the research method only has linear trend and correlation analysis, without highlight or innovative points, and the workload is not enough.
4. Strengthen the discussion on the correlation between temperature and precipitation and GPP. For example, what is the correlation between temperature, precipitation and GPP in growing season and non-growing season as described above, and the relationship between temperature and precipitation and GPP under the coupling influence of temperature and precipitation?
5. Add relevant diagrams to describe the conclusion in a more intuitive way.
6. Check for grammatical errors and improve your writing.
① The figure presents a single form, the composition is not coordinated, there is space for beautification.
② There are many mistakes in writing and grammar.
For example, Figure 2a in Line-197 should be changed to Figure 2.
Line-208 "The proportion of area with GPP of 100~150 g c/m2..." It has been described that 400-600 is the highest proportion, why add a sentence 100-150 is the highest proportion, and it does not correspond to the following statement.
Line-221~253: There are many mixed tenses in the grammar. The past tense is mixed with the present tense. Singular and plural errors, grammar errors. There are a lot of mistakes like this.
7. Select more representative references.
Author Response
Thank you very much for your valuable review. Manuscript is revised as your comments seriously. The revised part is highlighted with red color. Detailed response for each specific comments is attached.

Reviewer 2 Report
The manuscript entitled "Dynamics of Vegetation Productivity in Relation to Surface Meteorological Factors in Altay Mountain, Northwest China” aims at establishing a relationship between vegetation gross primary productivity (GPP) derived from MODIS data products and relevant meteorological factors in Altay Mountain; northwest China. I have read this manuscript with joy and I found it quite interesting. The problem being addressed is definitely of interest to a wide range of readership and the methodology is sound and well structured. In addition, the provided information in this study represents an important contribution. However, some changes are needed.
Comments and suggestions:
Introduction:
-Line 67: Please check “Zhou et al. analyzed”
- Line 70: “Wang Yingying” please check this.
- In the last section of the introduction the aims should be spelled out clearly please.
- Introduction requires a new section (few sentences) on the available techniques and other Satellite data products that are used in similar studies.
Materials and Methods
- Line 108: Please check “800 km2”
- Line 138: Firs time mentioning NDVI please use it properly in terms of citation and description
- Line 142-143: References are need for both” Savitzky Golay filtering” and the “MVC” synthesis please. Please justify why this type of filtering was selected. Then what is MVC?
Results and Discussion:
The discussion is not comprehensive and un-satisfactory; more is required on drawing parallels with other studies particularly recent studies. In addition, pinpointing the limitation of the study is also necessary.
Conclusion:
Please add another point highlighting the advantages of the used methodology and data products.
Author Response
Thank you very much for your valuable comments. The manuscript is revised as your comments seriously. The revised part is highlighted with red color. Detailed responses for each specific comments are attached.

Round 2
Reviewer 1 Report
Although the author has made some text revisions on the basis of the original manuscript. However, after reading it carefully, I still think that the biggest problem of this article is the lack of sparkle. It's a little flat, and it lacks the most compelling thing for readers.
Author Response
Thank you very much for your valuable comment. The Introduction and Conclusion section of manuscript are improved by adding relevant content.

Reviewer 2 Report
Dear Authors,
Thank you for addressing my concerns regarding the manuscript.
Author Response
Thank you very much for your valuable comment which will give us a great encouragement.
